# Quality Evaluation of Ready-to-Eat Coated Clementine (*Citrus x Clementina*) Fruits

Miriam Arianna Boninsegna [1], Alessandra De Bruno [2,*] and Amalia Piscopo [1]

[1]  Department AGRARIA, University Mediterranea of Reggio Calabria, Via dell'Università 25, 89124 Reggio Calabria, Italy; amalia.piscopo@unirc.it (A.P.)
[2]  Department of Human Sciences and Promotion of the Quality of Life, San Raffaele University, 00166 Rome, Italy
*   Correspondence: alessandra.debruno@uniroma5.it

**Abstract:** Conventional and innovative preservation treatments were compared to extend the shelf life of ready-to-eat Clementine (*Citrus x Clementina*) segments. The aim of this research was to find an environmentally friendly packaging typology for this fruit while preserving quality and meeting the needs of the consumer in terms of practicality of use and food safety. The experimental plan envisaged both the use of conventional storage techniques, such as modified atmosphere packaging ($O_2$ 5%, $CO_2$ 5%, and $N_2$ 90%), and the use of innovative storage techniques, such as an alginate-based (1.5%) edible coating. Quality changes were monitored by evaluating several indexes, such as color, texture, weight loss, respiration rate, pH, solid soluble content, bioactive compounds, antioxidant activity, organic acids, and microbiological contamination for 21 days at 4 °C. Moreover, a panel of judges assessed the sensory characteristics. Ready-to-eat Clementine segments, produced with edible coatings, possessed better sensory and textural properties and similar physic-chemical characteristics than those packaged in a modified atmosphere. The coating favored the creation of a controlled environment with low oxygen stress, which resulted in a reduction in enzymatic activity and oxidation for 20 days of storage at 4 °C. The results suggest that an edible coating could be a sustainable alternative to a modified atmosphere for the shelf life extension of ready-to-eat Clementine segments.

**Keywords:** *Citrus x Clementina*; coating; ready-to-eat





## 1. Introduction

In recent years, the increased focus on the health properties of foods and the fast-paced lifestyle of the modern consumer have led to an exponential increase in demand for ready-to-eat fruits.

The Clementine (*Citrus x Clementina*) is a citrus fruit. It is a hybrid between the Mediterranean mandarin and the sweet orange, whose maturation typically takes place in autumn; they have found their natural habitat in Calabria, a region in Southern Italy. Often, the Clementine is mistakenly confused with the mandarin, from which it is distinguished by its sweet taste, the absence of seeds (with rare exceptions), the ease of being peeled, a more intense orange color in the peel, and the absence of the characteristic scent of the mandarin. Very fragrant and sweet, Clementine fruits are eaten fresh or used for the preparation of syrups, juices, jams, and in many pastry recipes for the preparation of cakes and pies, or to obtain ice creams, sorbets, and jellies. This citrus fruit has been recognized by many authors as source of countless bioactive compounds with health-promoting properties for humans, such as vitamins (in particular, C), carotenoids, flavonoids, and phenolic acids [1,2]. However, post-harvest operations, such as peeling and cutting, can significantly affect the shelf life, as they favor metabolic processes that cause a sudden qualitative decay [3]. This type of vegetable processing in fact involves a faster decay of minimally processed fruits because it triggers a series of chained reactions that determine enzymatic browning,

softening, microbial contamination in vegetables tissues, and the final production of volatile substances [2–5]. These reactions drastically reduce the chemical, physical, and sensorial characteristics, as well as food safety. For these reasons, maintaining the qualitative parameters and delaying the growth of pathogen and spoilage microorganisms during storage are real challenges for the fresh fruit industry [3].

Nowadays, fruits are stored using low temperatures (equal to or less than 4 °C), often in combination with modified atmosphere packaging (MAP), which represent packages with unbalanced gaseous composition with respect to the normal atmospheric gas composition (low concentration of oxygen and high concentration of carbon dioxide) to counteract reactions that, assisted by the presence of high oxygen rates, lead to chemical and microbiological degradation of the fruit. Nevertheless, this method has some limitations due to the loss of its effect after opening or possible mechanical damage during transport/sale (holes, cuts, etc.) [6]. Edible coatings are a valid and environmentally sustainable technology to modify the atmosphere to extend the shelf life of ready-to-eat fruits [7,8]. Sodium alginate and a calcium chloride solution can be used to formulate the coating on the surface of vegetables [9]. In the presence of calcium bivalent ions ($Ca^{++}$), there is a phenomenon of molecular cross-linking that determines the strengthening of chemical bonds between the components of sodium alginate and promotes the barrier effect of migration of the coating water [8]. Several studies show that edible alginate-based coatings have the potential to supply a selective barrier to moisture, carbon dioxide, and oxygen, improve mechanical/textural properties, and prevent flavor loss [7–9]. In addition, it has been reported that edible coatings allow control of the processes of transpiration and respiration (which cause fast weight loss and fruit dissection), slow down enzymatic activity, and help to preserve the healthy characteristics of the fruit [3,7–11].

The replacement of the use of a modified atmosphere with edible coatings to extend the shelf life of perishable products could represent an eco-friendly choice as the materials used for the realization of the coatings are obtained from renewable sources [10–13].

The aim of this work was to test the effect of quite recent preservation methods (MAP) and innovative ones, such as an edible alginate-based coating, on keeping the chemical, physical, microbiological, and sensory characteristics of ready-to-eat Clementine segments. The quality change was monitored by evaluating several microbiological, sensory, and physic-chemical indexes during storage for 21 days at 4 °C.

Previous studies have evaluated the influence of edible coatings on whole citrus fruits. Alvarez et al. [14] saw that pectin-based coatings enriched with eugenol preserved the chemical-physical characteristics of 'Valencia' oranges and reduced the incidence of sour rot caused by *P. Italicum*. Jurić et al. [15] noted that layer-by-layer hydroxypropyl methylcellulose/chitosan or single chitosan coating preserved the overall quality of mandarin fruit both at room and cold temperatures for 10 and 28 days, respectively. Rasouli et al. [16] evidenced that an edible coating based on Aloe vera gel and salicylic acid reduced electrolyte leakage, chilling injury, and malondialdehyde accumulation and preserved sensorial, textural, and microbiological characteristics of the orange 'Thomson Navel.' While edible coatings on whole citrus fruits have already been tested [14–16], no coating has yet been tested on segments of Clementine to produce ready-to-eat fruits. In addition, there are no studies concerning the packaging and storage of Clementine segments, so this research could be useful for disseminating new knowledge on equally new possibilities for technological proposals and the marketing of quality fruit.

## 2. Materials and Methods

### 2.1. Reception and Pretreatment of Raw Material

The Clementine fruits (*Citrus x Clementina*) were bought at a local market and transported to the FoodTec laboratory of the University Mediterranean of Reggio Calabria, and those with defects (presence of mold, physical damage, parasitic attacks, etc.) were removed.

Citrus fruits with weights between 80–90 g, heights $\geq$ 50 mm, widths $\geq$ 60 mm, and external peel 'albedo' of a completely orange color were selected. Subsequently, to decrease the microbial contamination on the 'albedo,' the whole fruits were dipped in a solution of sodium hypochlorite (200 ppm) for 2 min, washed with distilled water, and dried on stainless steel grids in a vertical laminar flow hood (UV lamp 30 W, mod. ASALAIR 1200 FLV, Asal Srl, Milan, Italy) for 30 min at room temperature in forced air [16].

The preparation of the raw material and the coating of the Clementine segments were performed in a vertical laminar flow hood (UV lamp 30 W, mod. ASALAIR 1200 FLV, Asal Srl, Milan, Italy).

All tools used during preliminary and coating operations were sanitized before use with a solution of sodium hypochlorite (50 ppm).

### 2.1.1. Preparation Coating Solution

Food grade sodium alginate (Sigma-Aldrich, Merk Life Science s.r.l., Milano, Italy), glycerol (Carlo Erba reagents, Cornaredo, Italy), and calcium chloride (Labochimica s.r.l., Campodarsego, Italy) were used to prepare the solutions to create the edible coating on the surface of the Clementine segments.

The sodium alginate solution (1.5% $w/v$) was prepared by dissolving the sodium alginate powder in distilled water at 70 °C with magnetic stirring for 60 min. Then, the solution was cooled at room temperature (up to 30 °C), and glycerol (1.5% $w/v$) was added as a plasticizer to increase the coating flexibility. Calcium chloride solution (2% $w/v$) was prepared by dissolving calcium chloride in distilled water under magnetic stirring at room temperature for 30 min [17].

### 2.1.2. Treatment and Storage of Clementine Slices

Clementine fruits, after the treatment described in Section 2.1, were peeled, and the segments were separated manually. Subsequently, a part of these segments was coated with alginate solution (AL), and the remaining segments were directly packaged in a modified atmosphere (MAP) with $O_2$ 5%, $CO_2$ 5%, and $N_2$ 90% gas composition; the control sample (CTR) was packaged in a normal atmosphere.

Regarding the application of the edible coating on the surface of ready-to-eat Clementine fruits, the segments were dipped for 2 min in sodium alginate solution (1.5% $w/v$) and recovered, and the excess of the solution was air dried at room temperature for 1 min. Later, they were again immersed in calcium chloride solution (2% $w/v$) for 2 min to induce the cross-linked reaction, and they were air dried at room temperature up to complete drying [17].

Clementine segments (about 100 g) were packaged in a PP tray that was heat sealed with PP/PE film using a packaging machine (Orved, VGP 25N, Musile di Piave, Italy). The samples were stored at 4 °C for 21 days under constant lighting to simulate the real conditions of sale.

Physical, chemical, and microbiological analyses were carried out at 0, 3, 7, 14, and 21 days of storage. Sensory analyses were conducted at the beginning and end of storage. Figure 1 shows a schematic representation of this study on ready-to-eat Clementine segments.

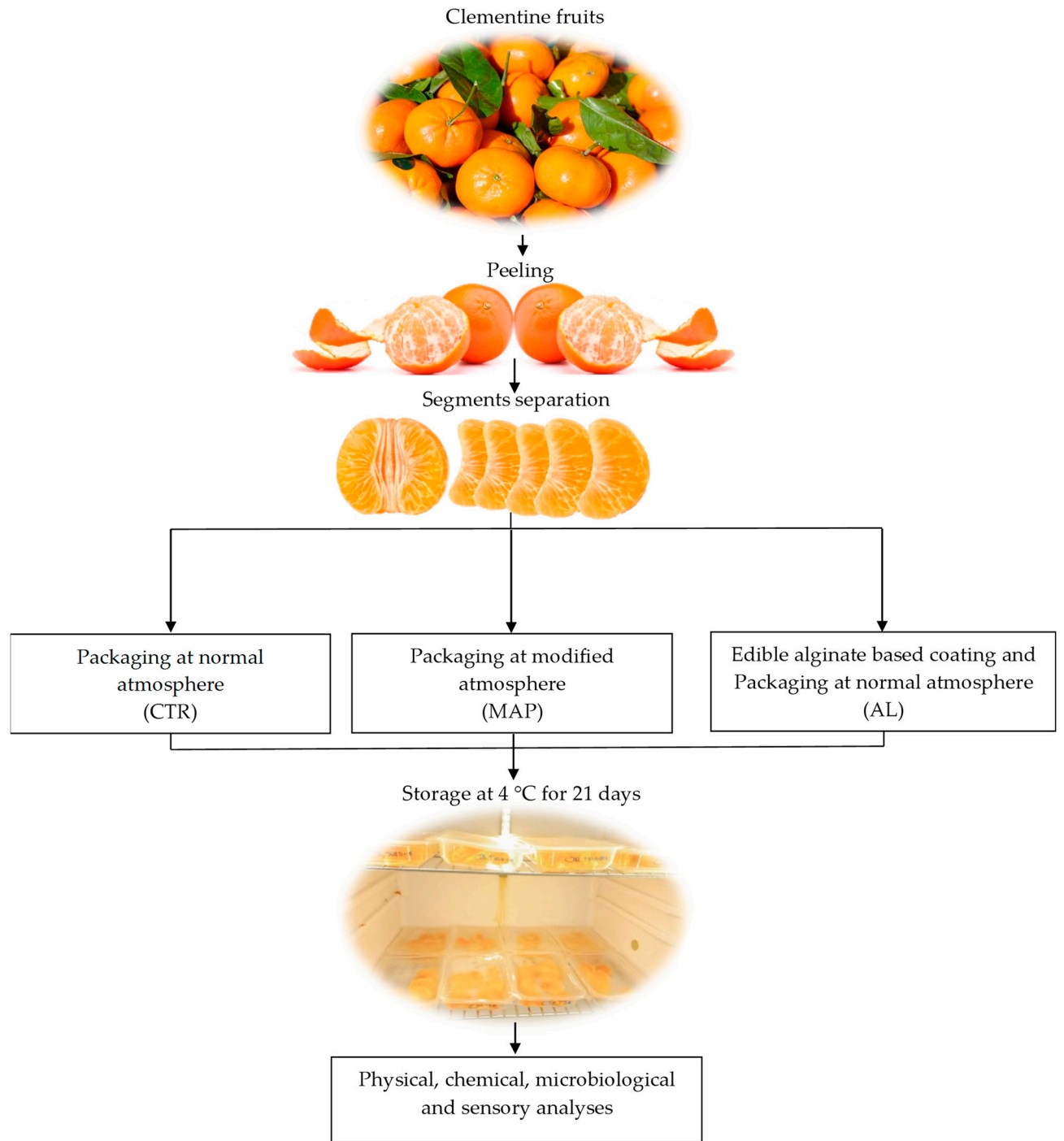

**Figure 1.** Schematic representation of this ready-to-eat Clementine segments study.

### 2.2. Color Measurement

The Clementine surface color parameters were determined according to the CIE L*a*b* color system by colorimeter (Minolta CM-700d Spectrophotometer, Konica Minolta, Inc., Sakai, Osaka, Japan) using a D65 illuminant. The L* parameter represents the lightness of the sample on the 0–100 scale, where 0 is the black and the 100 is the white. The positive a* represents the red content and the negative a* represents the green content of the sample on the red/green axis. The positive b* represents the yellow content and the negative b* represents the blue content of the sample on the yellow/blue axis. Color variables were recorded for each sample per treatment (12 segments × 2 replicates).

### 2.3. Weight Loss, Moisture, and Headspace Gas Composition

Weight loss was determined according with the AOAC standard method [18], expressed in percentage and calculated by the following Equation (1):

$$\text{Weight loss } (\%) = \frac{W_b - W_m}{W_b} \times 100 \tag{1}$$

where $W_b$ is the weight at the beginning of storage and $W_m$ is the weight monitored after 3, 7, 14, and 21 days of storage.

The moisture content was assessed gravimetrically by the AOAC method [19] as the difference in weight pre- and post-drying in the oven at 105 °C until a constant weight. The moisture content was expressed in percentage and calculated as follows (Equation (2)):

$$\text{Moisture content } (\%) = \frac{W_i - W_f}{W_i} \times 100 \tag{2}$$

where $W_i$ is the weight of the fresh samples and $W_f$ is the weight of the samples after drying.

The headspace composition was recorded using a gas analyzer (PBI, DANSENSOR, Ringsted, Denmark, CP $O_2/CO_2$) equipped with a needle to withdraw and analyze the gaseous composition of the tray's headspace. The needle was inserted into the tray using a neoprene plastic pad to prevent gases from the surrounding atmosphere from entering during measurement.

### 2.4. Textural Analysis

The textural analysis was conducted with a penetration test to determine the hardness of the Clementine segments according to the method proposed by Glicerina et al. [17] using a TA-XT Plus Texture Analyzer (Stable Micro Systems Ltd., Godalming, UK) equipped with a 5 mm diameter stainless steel probe (P/5). Data acquisition and curve integration were carried out using Exponent software 6.1.4.0 (Stable Micro Systems Ltd., Godalming, UK). The parameters used for this test were: penetration distance of 3 mm, test speed of 1.0 mm/s, and post-test speed of 3.0 mm/s.

The hardness was expressed in grams (g) and estimated as the maximum peak of the curve recorded during the penetration test (the highest force necessary to penetrate the entire segment). Twenty replicates were used for each sample.

### 2.5. Sensory Analysis

A sensory quantitative descriptive analysis (QDA) was performed to assess the sensorial attributes of ready-to-eat Clementine segments.

Ten participants (aged 21–42) with earlier sensory analysis experiences were recruited to evaluate the visual, gustatory, olfactory, and structural attributes of the fruits. The sensory analysis was based on a 0-to-9-point hedonic scale where 0 showed the absence of the attribute and 9 showed an extremely high attribute value. The acceptability limit was considered to be 4.5. The visual appearance (intensity of the color, form, glossiness, uniformity of the surface), aroma (intensity, fruity, citrus, spicy), taste (sweet, salt, acid, bitter, citrus, fruity, astringent, aftertaste), texture descriptors (consistency, chewiness, moisture, crunchiness, turgidity), and total acceptability were evaluated. The results were expressed as an average of the judgements obtained during the tasting.

### 2.6. Chemical Analysis

About 70 g of Clementine segments was homogenized using an Ultra-Turrax (T 25 digital, IKA, Staufen, Germany) and then transferred in a falcon tube and centrifuged (NF 1200R, Nüve, Ankara, Turkey) for 10 min at 10,000 rpm at 4 °C. The supernatant was recovered, filtered through a Buchner apparatus with a 0.45 mm filter paper, and filtered again with a PTFE 0.45 μm (diameter of 15 mm) syringe filter. The obtained juice was used

to determine the pH, total soluble solids (TSS), titratable acidity (TA), total phenolic content (TPC), total flavonoid content (TFC), and total antioxidant activity (TAA).

### 2.6.1. Total Soluble Solids (TSS) and Titratable Acidity (TA)

The total soluble solids were estimated by placing a few drops of juice on the prism of a digital hand refractometer (DBR 047 SALT, Giorgio Bormac s.r.l, Carpi (MO), Italy) and expressed in degrees Brix (°Bx) at 25 °C.

As regards the titratable acidity, 5 mL of juice was diluted with 50 mL of deionized water and titrated with 0.1 M NaOH until pH 8.1 using a digital pH meter (Crison Basic 20, Crison instruments, Alella, Spain). The results were expressed as g citric acid/100 g, as the most abundant acid in citrus [20].

### 2.6.2. Total Phenolic Content (TPC)

The TPC was determined with a colorimetric method described by Jurić S. et al. [12], with some modifications. Briefly, 0.1 mL of juice, 5 mL of distilled water, and 0.5 mL of Folin-Ciocalteau reagent (diluted in 1:2 ratios with distilled water) were mixed in a 10 mL volumetric flask. After 2 min, 1.5 mL of 20% sodium carbonate solution ($v/v$) was added. The reaction mixture was made up to volume with distilled water and kept in the dark room for 2 h at room temperature. The solution used as a blank was prepared by replacing the sample with water in the reaction mixture. The absorbance was recorded at 765 nm against a blank using a spectrophotometer (Perkin-Elmer UV-Vis k2, PerkinElmer Inc., Waltham, MA, USA) and by comparing with a gallic acid calibration curve (1–10 mg L$^{-1}$). The results were expressed as mg of gallic acid equivalents/kg of fresh weight.

### 2.6.3. Total Flavonoid Content (TFC)

The TFC was carried out with a modified method proposed by Jurić S. et al. [15]. In a 5 mL volumetric flask were mixed 0.2 mL of juice, 2 mL of distilled water, and 0.15 mL of 5% of sodium nitrite ($v/v$). After 5 min, 0.15 mL of 10% aluminum chloride ($v/v$) was added to the reaction mixture, which was then incubated at room temperature for 6 min. Subsequently, 1 mL of 1 M sodium hydroxide was added, made up to volume with distilled water, and kept at room temperature for 10 min. A blank solution was prepared by replacing the sample with water in the reaction mixture. The absorbance was recorded at 360 nm against a blank using a spectrophotometer (Perkin-Elmer UV-Vis k2, PerkinElmer Inc., Waltham, MA, USA) and by comparing with a quercetin calibration curve (1–20 mg L$^{-1}$). The results were expressed as mg of quercetin equivalents/g of fresh weight.

### 2.6.4. Determination of Organic Acids

For the determination of organic acids (oxalic, malic, ascorbic, and citric acids) in Clementine samples, HPLC (High-performance liquid chromatography) analysis was used according to the literature [15]. Clementine juices were diluted when necessary and filtered by a PTFE 0.45 μm (diameter of 15 mm) syringe filter before analysis.

A Knauer HPLC Smartline Pump 1000, equipped with a Knauer Smartline UV Detector 2600 and a thermostat, was used. The separation of organic acids was performed on SYNERGI HYDRO-RP (250 mm × 4.6 mm i.d., 4 μm) at 22 °C. The injected sample volume was 20 μL. The analysis was carried out in isocratic elution at a flow rate of 0.7 mL min with potassium phosphate 20 mM at pH 2.9 as the mobile phase. The detection wavelengths were 254 nm for ascorbic acid and 210 nm for malic, oxalic, and citric acid. The results are expressed as mg of acid/L of juice.

### 2.6.5. Total Antioxidant Activity (TAA)

The antioxidant activity of Clementine segments was measured using the ABTS and DPPH assays following the procedures proposed by De Bruno et al. [21], appropriately modified.

Concerning the DPPH assay, a $6 \times 10^{-5}$ methanolic solution of 2,2-diphenyl-1-picrylhydrazyl (DPPH) was prepared. The analysis was carried out by reacting in a cuvette 20 μL of Clementine juice at 2980 μL of methanolic DPPH radical solution. After 30 min of incubation in the dark at room temperature, the decrease in absorbance to 515 nm (proportional to the radical scavenging activity of the sample) was recorded using a spectrophotometer (Perkin-Elmer UV-Vis k2, PerkinElmer Inc., Waltham, MA, USA). Methanol was used as a blank.

Regarding the ABTS assay, the preparation of ABTS solution involved the reaction of 7 mM of ABTS (2,2-Azino-bis 3-ethylbenzothiazoline-6-sulfonic acid) solution and 2.45 of mM potassium persulfate. The mixture was kept in the dark for 12–16 h at room temperature. Then, the ABTS radical solution was diluted with ethanol up to 0.7 of absorbance at 734 nm, determined spectrophotometrically (Perkin-Elmer UV-Vis k2, PerkinElmer Inc., Waltham, MA, USA). The analysis of the samples was performed by mixing 20 μL of Clementine juice at 2980 μL of methanolic ABTS solution. The decrease in absorbance was read after 6 min of dark incubation at room temperature. Ethanol was used as a blank.

The results of both the DPPH and ABTS assays were expressed as mM Trolox equivalent $kg^{-1}$ of fresh Clementine fruits plotted against the Trolox concentration (from 1 to 24 μM).

### 2.7. Microbiological Analysis

The total bacterial count (TBC) and yeasts and molds (Y&M) were detected to evaluate the microbiological contamination of Clementine segments following Glicerina et al. [17], with some modifications. For the microbial analysis, 5 g of each sample was placed in a sterile bag with a Ringer solution and homogenate using Stomacher (BagMixer® 400 P, Interscience, Saint-Nom-la-Bretèche, France) for 3 min. The obtained samples were serially diluted, and 1 mL of each dilution was used for microbiological analysis. Dichloran Rose Bengal Chloramphenicol (DRBC) was used for the Y&M and the TBC. The plates, after solidification, were incubated at 25 °C, and colonies enumeration was made after 5 days and after 2 days for the Y&M and TBC counts, respectively. The results were expressed as log10 CFU/g of sample.

### 2.8. Statistical Analysis

The analytical data were reported as the mean value ± standard deviation. The analysis of variance (one-way ANOVA) was conducted by SPSS Software (Version 15.0, SPSS Inc., Chicago, IL, USA) by applying the Tukey post hoc test at $p < 0.05$.

## 3. Results

### 3.1. Color Measurements

The color of food is a very important index of quality for the consumer.

The results reported in Table 1 evidence no significant differences for L* and b* parameters among the samples and during storage times, whereas some light variations of the red color only appeared after 3 days. At the end of storage, all of the color parameters were similar among the Clementine samples.

**Table 1.** Colorimetric coordinates of Clementine samples during storage.

| Parameter | Sample | Time (Days) | | | | | Sig. |
|---|---|---|---|---|---|---|---|
| | | 0 | 3 | 7 | 14 | 21 | |
| L* | CTRL | 50.19 ± 1.69 | 50.72 ± 1.42 | 53.25 ± 1.22 | 52.86 ± 2.65 | 51.71 ± 1.67 | n.s. |
| | MAP | 48.97 ± 1.84 | 50.94 ± 1.54 | 52.31 ± 1.98 | 50.68 ± 2.03 | 52.51 ± 1.88 | n.s. |
| | AL | 54.80 ± 1.33 | 52.62 ± 1.69 | 53.84 ± 1.16 | 52.57 ± 1.54 | 53.92 ± 2.10 | n.s. |

**Table 1.** *Cont.*

| Parameter | Sample | Time (Days) | | | | | Sig. |
|---|---|---|---|---|---|---|---|
| | | **0** | **3** | **7** | **14** | **21** | |
| Sign. | | n.s. | n.s. | n.s. | n.s. | n.s. | |
| a* | CTRL | 8.30 ± 1.21 [b] | 8.41 ± 1.27 [b] | 8.74 ± 1.01 | 8.85 ± 2.14 | 8.67 ± 1.32 | n.s. |
| | MAP | 9.73 ± 1.00 [a] | 8.35 ± 1.58 [b] | 8.42 ± 1.18 | 8.43 ± 1.14 | 9.25 ± 1.48 | n.s. |
| | AL | 8.18 ± 1.3 [b] | 9.27 + 1.01 [a] | 8.97 ± 1.53 | 8.60 ± 1.96 | 9.46 ± 1.56 | n.s. |
| Sign. | | * | * | n.s. | n.s. | n.s. | |
| b* | CTRL | 19.48 ± 1.07 | 20.90 ± 1.54 | 19.78 ± 1.43 | 21.71 ± 97 | 20.21 ± 1.82 | n.s. |
| | MAP | 19.66 ± 1.21 | 19.92 ± 1.36 | 21.23 ± 1.61 | 21.90 ± 2.15 | 21.83 ± 1.59 | n.s. |
| | AL | 20.27 ± 1.5 | 20.48 ± 1.41 | 21.63 ± 1.53 | 20.60 ± 2.63 | 21.39 ± 1.58 | n.s. |
| Sign. | | n.s. | n.s. | n.s. | n.s. | n.s. | |

Small letters within a column show significant differences as assessed by Tukey's post hoc test. *, significance at $p < 0.05$; n.s., not significant.

### 3.2. Moisture, Weight Loss, and Headspace Gas Composition

The trends of moisture, weight loss, and headspace gas composition are shown in Table 2. All samples showed very light weight loss with values equal to 0.1% after 21 days of storage. In fact, similar moisture contents (85%–86%) were observed in all samples.

**Table 2.** Moisture, weight loss, and headspace gas composition of Clementine segments during storage.

| Parameter | Sample | Time (Days) | | | | | Sig. |
|---|---|---|---|---|---|---|---|
| | | **0** | **3** | **7** | **14** | **21** | |
| Moisture (g/100 g) | CTRL | 85.02 ± 0.55 [B] | 88.46 ± 2.00 [B] | 87.60 ± 1.90 [AB] | 83.2 ± 2.16 [A] | 85.59 ± 1.85 [AB] | * |
| | MAP | 85.65 ± 1.50 | 86.10 ± 1.50 | 82.93 ± 1.16 | 83.3 ± 1.66 | 86.69 ± 0.98 | n.s. |
| | AL | 85.38 ± 2.00 | 85.56 ± 0.30 | 86.74 ± 1.97 | 85.8 ± 1.35 | 86.77 ± 1.23 | n.s. |
| Sign. | | n.s. | n.s. | n.s. | n.s. | n.s. | |
| Weight Loss (g/100 g) | CTRL | - | 0.03 ± 0.01 [A] | 0.02 ± 0.01 [A] | 0.07 ± 0.02 [B] | 0.10 ± 0.03 [B] | ** |
| | MAP | - | 0.03 ± 0.01 [A] | 0.02 ± 0.01 [A] | 0.08 ± 0.02 [B] | 0.10 ± 0.03 [B] | ** |
| | AL | - | 0.05 ± 0.02 [A] | 0.05 ± 0.01 [A] | 0.07 ± 0.01 [AB] | 0.10 ± 0.04 [B] | ** |
| Sign. | | | n.s. | n.s. | n.s. | n.s. | |
| $O_2$ (%) | CTRL | 21.00 ± 0.00 [aA] | 14.5 ± 1.11 [aBC] | 14.3 ± 1.90 [aBC] | 7.1 ± 0.99 [bB] | 5.6 ± 0.71 [bC] | ** |
| | MAP | 5.00 ± 0.00 [bA] | 3.4 ± 0.98 [bAB] | 1.8 ± 0.40 [bA] | 1.2 ± 0.43 [cA] | 1.1 ± 0.05 [cA] | ** |
| | AL | 21.00 ± 0.00 [aA] | 16.4 ± 0.43 [aB] | 13.3 ± 0.63 [aBC] | 8.25 ± 0.57 [aC] | 8.3 ± 0.68 [aC] | ** |
| Sign. | | ** | ** | ** | ** | ** | |
| $CO_2$ (%) | CTRL | 0.02 [bC] | 9.4 ± 1.37 [B] | 10.0 ± 2.30 [bB] | 13.0 ± 1.13 [bB] | 19.9 ± 0.14 [aA] | ** |
| | MAP | 5.00 [aC] | 9.9 ± 0.76 [B] | 13.5 ± 1.56 [aBC] | 17.2 ± 2.3 [aA] | 15.8 ± 1.79 [bBC] | ** |
| | AL | 0.02 [bC] | 9.2 ± 0.43 [B] | 9.8 ± 0.81 [bB] | 14.9 ± 0.10 [abA] | 14.3 ± 0.49 [bA] | ** |
| Sign. | | ** | n.s. | * | ** | * | |

Small letters within a column and capital letters within a row show significant differences as assessed by Tukey's post hoc test. **, significance at $p < 0.01$; *, significance at $p < 0.05$; n.s., not significant.

The results of gas composition denoted the effect of fruit metabolism during storage in a different trend, depending on the applied technology. The oxygen decreased during monitoring times from 21% to 5.6% (CTR) or from 8.3% (AL) and 5% to 1.1% (MAP), with a consequent increase in the carbon dioxide percentage (14.3%–19.9%).

### 3.3. Textural and Sensory Analysis

As illustrated in Figure 2, a general increase in hardening was observed in all samples during the storage time, whereas the samples showed significant differences. The coated segments (AL) possessed a higher firmness (279.3 g) than the MAP (164.9 g) and

CTR (206.2 g). This trend was probably due to reactions that cause tissue breakdown or thickening.

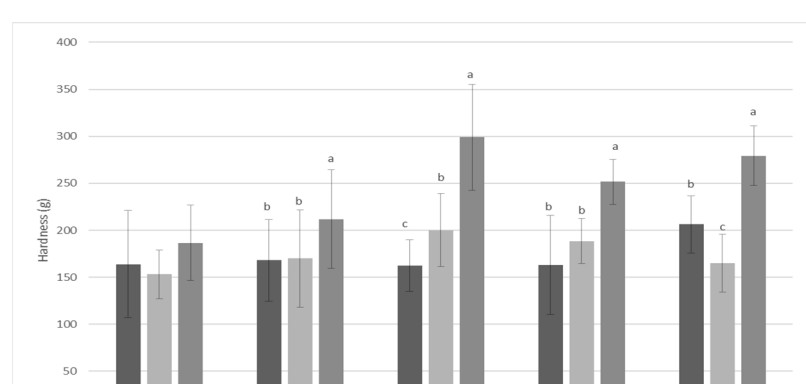

**Figure 2.** Textural proprieties of Clementine segments during storage. Different letters show significant differences between each group as assessed by Tukey's post hoc test ($p < 0.05$).

The sensory analysis of samples (Table 3) showed a lower score for sweet (5.5), fruity (5), and citrusy (5.3), while a higher score was given for bitter (6.2) in AL than CTR and MAP (around 6.7 for positive gustatory hints and, respectively, 5.7 and 4.5 for negative ones), among the gustatory hints. However, the judgements about the visual and structural characteristics indicated that AL were the most appreciated Clementine samples by panelists at up to 20 days of storage (color score: 8; turgidity score: 7). Finally, regarding the total acceptability, the coated Clementine segments obtained a higher score than MAP and CTR after 21 days of storage at 4 °C (6.2).

**Table 3.** Sensory parameters of Clementine segments during storage.

| Parameter | Sample | Time (Days) | | Sig. |
| --- | --- | --- | --- | --- |
| | | **0** | **21** | |
| Sweet | CTRL | 6.80 ± 1.30 [a] | 4.00 ± 0.08 [b] | ** |
| | MAP | 6.70 ± 1.20 [a] | 5.00 ± 0.60 [a] | ** |
| | AL | 5.50 ± 1.00 [b] | 5.00 ± 0.80 [a] | * |
| Sign. | | * | * | |
| Bitter | CTRL | 5.70 ± 0.90 [ab] | 6.00 ± 1.20 [ab] | n.s. |
| | MAP | 4.50 ± 1.00 [b] | 4.00 ± 1.30 [a] | n.s. |
| | AL | 6.20 ± 0.70 [a] | 7.00 ± 1.20 [b] | n.s. |
| Sign. | | * | ** | |
| Fruity | CTRL | 6.70 ± 0.70 [a] | 6.00 ± 1.00 | n.s. |
| | MAP | 6.70 ± 0.70 [a] | 6.00 ± 1.50 | n.s. |
| | AL | 5.00 ± 0.80 [b] | 6.00 ± 0.80 | * |
| Sign. | | * | n.s. | |
| Citrusy | CTRL | 6.70 ± 0.70 [a] | 6.00 ± 1.20 [ab] | n.s. |
| | MAP | 6.5 ± 0.50 [a] | 5.00 ± 0.80 [a] | ** |
| | AL | 5.30 ± 1.20 [b] | 7.00 ± 0.60 [b] | * |
| Sign. | | * | ** | |
| Color | CTRL | 7.00 ± 0.06 [b] | 6.50 ± 1.30 [b] | n.s. |
| | MAP | 6.50 ± 0.05 [b] | 6.00 ± 0.60 [b] | n.s. |
| | AL | 9.00 ± 0.05 [a] | 8.00 ± 0.60 [a] | n.s. |

**Table 3.** *Cont.*

| Parameter | Sample | Time (Days) 0 | Time (Days) 21 | Sig. |
|---|---|---|---|---|
| Sign. | | ** | * | |
| Turgidity | CTRL | 6.8 ± 1.1 a | 6.2 ± 0.9 a | n.s. |
| | MAP | 7.2 ± 1.1 ab | 6 ± 1.00 a | n.s. |
| | AL | 8.0 ± 0.6 b | 7 ± 1.2 b | n.s. |
| | | * | * | |
| Overallacceptability | CTRL | 8.00 ± 0.50 a | 5.00 ± 0.50 b | ** |
| | MAP | 8.00 ± 1.00 a | 6.00 ± 0.50 a | ** |
| | AL | 7.2 ± 0.50 b | 6.2 ± 0.50 a | ** |
| Sign. | | * | * | |

For letters, **, *, n.s. see Table 2.

### 3.4. Total Soluble Solids and Titratable Acidity

The total soluble solids (TSS) and titratable acidity (TA) are important quality indices that show the freshness of Clementine fruits and that are closely related to the sweet taste that makes them particularly appreciated by the final consumer. In this study, significant differences among the tested samples were noted after 7 days of storage. The results reported in Table 4 show the trend in all samples during storage.

**Table 4.** Total soluble solids and titratable acidity of Clementine segments during storage.

| Parameter | Sample | Time (Days) 0 | Time (Days) 3 | Time (Days) 7 | Time (Days) 14 | Time (Days) 21 | Sig. |
|---|---|---|---|---|---|---|---|
| SSC (°Bx) | CTRL | 12.1 ± 0.05 AB | 12.2 ± 0.17 aA | 10.0 ± 0.72 bC | 11.6 ± 0.1 aAB | 11.5 ± 0.38 B | ** |
| | MAP | 11.9 ± 0.4 A | 12.8 ± 0.1 aA | 11.5 ± 0.71 aA | 11.8 ± 0.07 bA | 12.3 ± 0.98 A | n.s. |
| | AL | 11.9 ±0.13 A | 12.2 ± 0.23 aA | 12.6 ± 0.78 aA | 12.2 ± 0.07 cA | 11.9 ± 0.06 A | n.s. |
| Sig. | | n.s. | n.s. | * | ** | n.s. | |
| Titratable Acidity (%) | CTRL | 0.59 ± 0.03 aA | 0.54 ± 0.02 aA | 0.53 ± 0.04 aA | 0.50 ± 0.05 aB | 0.51 ±0.06 aAB | * |
| | MAP | 0.57 ± 0.1 a | 0.57 ± 0.08 abA | 0.54 ± 0.08 aA | 0.52 ± 0.01 aA | 0.52 ± 0.05 aA | n.s. |
| | AL | 0.54 ± 0.03 aA | 0.54 ± 0.01 bA | 0.56 ± 0.05 aA | 0.54 ±0.03 aA | 0.54 ±0.01 aA | n.s. |
| Sign. | | n.s. | * | n.s. | n.s. | n.s. | |

For letters, **, *, n.s. see Table 2.

As regards the Clementine segments coated with alginate, a constant trend was recorded until the end of storage (11.9 °Bx and 0.54% for TSS and TA, respectively). Similar results were observed for MAP samples with parameters from 11.9 to 12.3 °Bx and from 0.57% to 0.52%, from the beginning until 20 days of storage.

In contrast, the CTR samples showed a statistically significant variation in these parameters due to the physiological mechanisms of the fruits following the deprivation of the albedo.

### 3.5. Total Phenolic Content (TPC), Total Flavonoid Content (TFC), Organic Acid, and Total Antioxidant Activity (TAA)

The healthy properties of Clementine fruits are due to a series of valuable compounds including polyphenols, flavonoids, and organic acids, whose variations observed in this study are reported in Figure 3 and Table 5. During the first days of storage, there was an increase in TPC and TFC for all of the analyzed samples. This phenomenon is due to enzymatic activities that occur immediately after the peeling of the fruits. A significant increase was found after 7 and 14 days of storage. However, on the 21st day of shelf life, the AL segments showed, significantly, the highest antioxidant contents compared to MAP and CTR (571.80 mg GAE kg$^{-1}$ and 432.85 mg QE kg$^{-1}$).

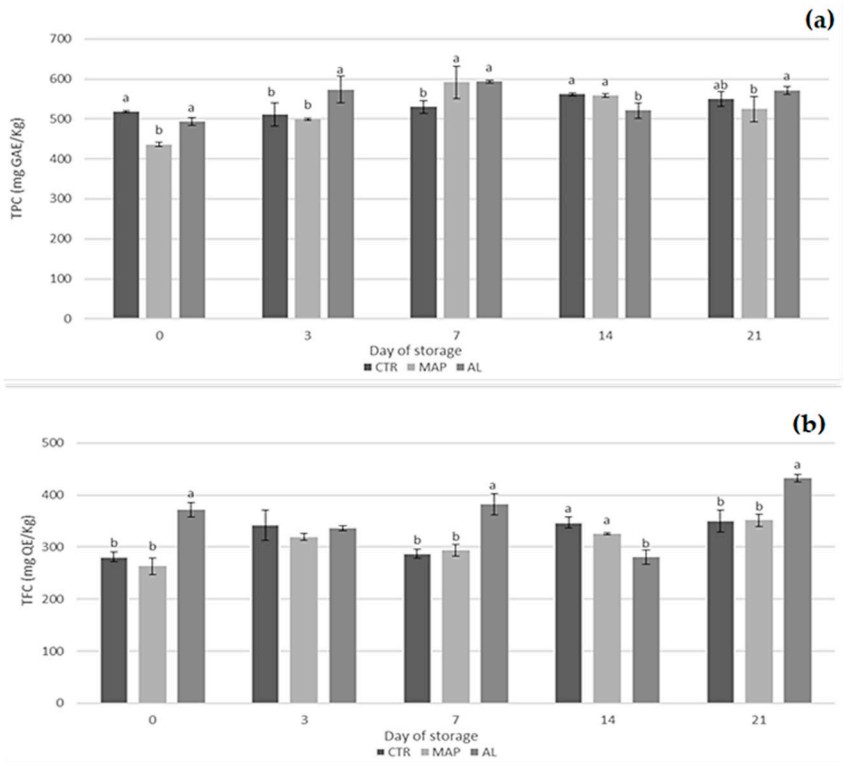

**Figure 3.** Total phenolic content (TPC) (**a**) and total flavonoid content (TFC) (**b**) in Clementine segments during storage. Different letters (**a**,**b**) show significant differences between the means of each group as assessed by Tukey's post hoc test ($p < 0.05$).

**Table 5.** Organic acid and total antioxidant activity (TAA) of Clementine segments during storage.

| Parameter | Sample | Time (Days) | | | | | Sig. |
|---|---|---|---|---|---|---|---|
| | | 0 | 3 | 7 | 14 | 21 | |
| Oxalic Acid (mg L⁻¹) | CTRL | 190.24 ± 5.70 [A] | 150.59 ± 2.70 [B] | 161.88 ± 2.70 [B] | 152.86 ± 1.60 [B] | 204.30 ± 2.80 [A] | ** |
| | MAP | 227.37 ± 8.20 [A] | 152.11 ± 9.40 [B] | 247.45 ± 7.20 [A] | 118.81 ± 1.80 [B] | 208.89 ± 2.80 [AB] | ** |
| | AL | 150.58 ± 6.30 | 147.59 ± 5.30 | 186.93 ± 5.80 | 186.24 ± 90 | 211.57 ± 12.40 | n.s. |
| | Sign. | n.s. | n.s. | n.s. | n.s. | n.s. | |
| Citric Acid (mg L⁻¹) | CTRL | 4818.34 ± 83.40 | 4478.34 ± 28.70 | 4925.01 ± 96.70 | 4430.29 ± 43.40 | 4489.26 ± 41.20 | n.s. |
| | MAP | 5502.17 ± 39.70 | 5061.70 ± 47.00 | 4367.87 ± 48.90 | 4996.61 ± 72.80 | 4444.42 ± 76.30 | n.s. |
| | AL | 5005.19 ± 20.70 | 4235.22 ± 59.40 | 4997.44 ± 43.40 | 4632.13 ± 72.00 | 4860.58 ± 64.50 | n.s. |
| | Sign. | n.s. | n.s. | n.s. | n.s. | n.s. | |
| Malic Acid (mg L⁻¹) | CTRL | 861.98 ± 42.80 [A] | 825.72 ± 62.80 [A] | 662.36 ± 54.00 [abAB] | 538.78 ± 42.90 [B] | 681.32 ± 44.70 [AB] | ** |
| | MAP | 615.39 ± 38.10 | 625.48 ± 58.00 | 504.96 ± 46.10 [b] | 481.06 ± 51.90 | 498.81 ± 58.70 | n.s. |
| | AL | 624.55 ± 54.30 [AB] | 713.00 ± 49.00 | 716.35 ± 68.70 [aA] | 483.06 ± 61.40 [B] | 625.54 ± 67.10 [AB] | * |
| | Sign. | n.s. | n.s. | * | n.s. | n.s. | |
| Ascorbic Acid (mg L⁻¹) | CTRL | 153.64 ± 19.25 [A] | 101.23 ± 25.30 [bAB] | 89.04 ± 5.88 [cB] | 97.19 ± 1.65 [bAB] | 90.46 ± 1.00 [bB] | ** |
| | MAP | 118.74 ± 6.10 | 110.71 ± 4.30 [ab] | 133.20 ± 0.50 [b] | 121.99 ± 22.30 [ab] | 127.93 ± 23.60 [ab] | n.s. |
| | AL | 124.49 ± 18.80 [B] | 147.18 ± 35.50 [aAB] | 169.82 ± 14.10 [aA] | 168.56 ± 4.70 [aA] | 167.85 ± 8.40 [aA] | ** |
| | Sign. | n.s. | * | ** | * | * | |
| ABTS (mM Trolox kg⁻¹) | CTRL | 1.33 ±0.20 [aB] | 1.80 ±0.05 [aA] | 1.31 ± 0.23 [aB] | 1.24 ± 0.05 [aB] | 1.42 ± 0.06 [bB] | ** |
| | MAP | 1.05 ±0.08 [bA] | 1.21 ±0.25 [bA] | 1.46 ± 0.35 [aA] | 1.30 ± 0.04 [aA] | 1.39 ±0.12 [bA] | n.s. |
| | AL | 1.26 ±0.12 [abB] | 1.97 ± 0.15 [aA] | 1.27 ± 0.17 [aB] | 1.32 ± 0.08 [aB] | 1.76 ±0.17 [aA] | ** |
| | Sign. | * | ** | n.s. | n.s. | ** | |
| DPPH (mM Trolox kg⁻¹) | CTRL | 0.43 ± 0.01 [aA] | 0.39 ± 0.02 [cAB] | 0.39 ± 0.02 [abAB] | 0.32 ± 0.01 [cB] | 0.40 ± 0.08 [aAB] | * |
| | MAP | 0.44 ± 0.05 [aAB] | 0.47 ± 0.01 [aA] | 0.31 ± 0.07 [bB] | 0.36 ± 0.02 [bAB] | 0.43 ± 0.1 [aAB] | * |
| | AL | 0.43 ± 0.04 [aB] | 0.44 ± 0.01 [bB] | 0.41 ± 0.01 [aB] | 0.41 ± 0.01 [aB] | 0.50 ± 0.03 [aA] | ** |
| | Sign. | n.s. | ** | * | ** | n.s. | |

For letters, **, *, n.s. see Table 2.

As regards the organic acid content (Table 5), both MAP and AL showed beneficial effects on the maintenance of citric, ascorbic, malic, and oxalic acids during the product's shelf life. Instead, in CTR samples, a drastic decrease was observed, especially in terms of ascorbic acid after three days of storage (from 153.64 to 101.23 mg $L^{-1}$).

The antioxidant activity recognized in citrus is due not only to the presence of phenolic compounds but also to the presence of organic acids. In this study, the synergy between the compounds mentioned above in counteracting the activity of free radicals was particularly evident from the results obtained from the ABTS and DPPH assays. In particular, the AL segments showed a gradual increase in radical scavenging activity from 1.26 to 1.76 and from 0.43 to 0.50 mM Trolox $kg^{-1}$, respectively. This was probably related to the previously described maintenance of the concentration of organic acids and the increase in TPC and TFC over time.

### 3.6. Microbiological Analyses

Microbiological analysis did not detect the presence of total microbial charge, yeast, or molds.

## 4. Discussion

Maintaining of the quality of ready-to-eat fruits over time is a real challenge for the fresh fruit industry. Mechanical operations, such as cutting or peeling, cause stress on the fruit, which can increase metabolic activity, exposure to microbiological contamination, and activation of enzymatic pools responsible for adverse effects (e.g., discoloration, loss of turgidity, and loss of nutritional proprieties). In this study, the quality of Clementine segments was studied in relation to two typologies of packaging, monitoring chemicals, and physical, microbiological, and sensory parameters during 21 days of storage at 4 °C.

The observed trend for moisture and weight loss was probably due to the specific characteristics of the packaging conditions and the metabolic activity of segments after peeling. In fact, the high barrier properties of the packaging prevented the diffusion of the aeriform from the inside to the outside of the tray. However, it should be noted that in the trays containing the CTR samples, an excess of fog was found during storage, indicating an intense metabolic activity due to transpiration and respiration processes accelerated by the deprivation of the peel from the fruit (Figure 4). This was also confirmed by the concentration of oxygen and carbon dioxide in the headspace of the trays during storage.

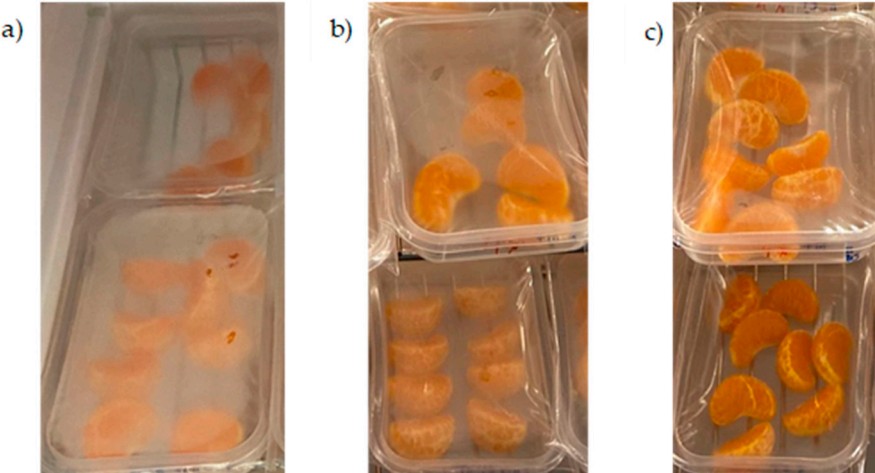

**Figure 4.** Ready-to-eat Clementine segments packaged at normal atmosphere (CTR) (**a**), at modified atmosphere (MAP) (**b**), and edible-alginate coated (**c**).

Comparing the oxygen and carbon dioxide levels of AL and CTR, it can be inferred that the coating applied on the segments slowed down their metabolic process.

The sensory characteristics, as well as the color and texture, play a key role in the choice of purchase by consumers. However, they have still been poorly studied. Previously, it was reported that the typical sweet taste of Clementine fruits can be strongly influenced by the constituent variables of the fruit (ratio of organic acid and solid soluble content), and also by the external stresses to which they are subjected pre/at harvest (cultivar, environmental condition, maturity/harvest period, physical damage, parasitic attacks, mold, etc.) and post-harvest (temperature, ratio of $O_2/CO_2$, permeability of packaging, microbiological attack, enzymatic reactions of degradation) [15,22,23]. Moreover, various volatile compounds, such as limonene, myrcene, $\alpha$-pinene, and linalool, are decisive in giving the typical odor and taste to the fruits of Clementine and mandarin [24–27].

The sensory evaluations revealed that the parameters related to the gustatory sensations were slightly influenced by the application of the coating at the end of storage. These results agree with earlier studies carried out on segments of oranges coated with alginate. It has been found that the calcium chloride used to promote the cross-linking of sodium alginate causes a greater feeling of bitterness in the fruit [17]. However, the parameters related to structure and visual appearance were significantly improved in AL samples.

The color and texture results confirmed the observations obtained through the panel test. Several studies show that edible alginate-based coatings can improve physical properties, such as mechanical properties, through cross-linking reactions between the structural components of the vegetable cell wall and $Ca^{++}$ ions, and visual ones, through retaining the color of the fruits as unchanged and bright [8]. Figure 5 shows the visual appearance of the Clementine segments after 21 days of storage.

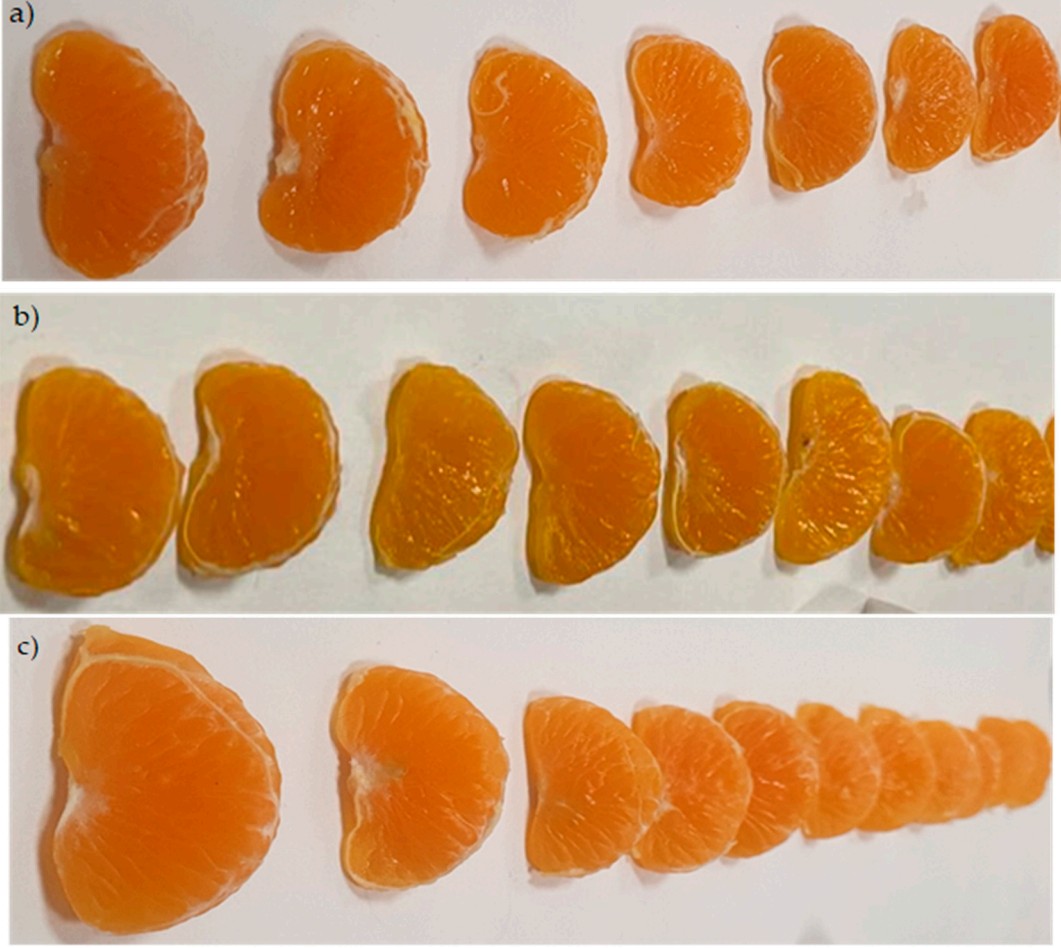

**Figure 5.** Visual appearance of Clementine segments after 21 days of storage. Segments at normal atmosphere (CTR) (**a**), packaged in modified atmosphere (MAP) (**b**), and edible-alginate coated (**c**).

As regards the chemical analyses, the TSS and TA levels in the juice bags are undoubtedly among the most important quality indices for Clementine fruits. During ripening, there is an increase in TSS and a decrease in TA that give citrus fruits the typical sweet taste [28,29]. The optimal maturity values of TSS and TA should be between 10 and 13 °Bx and from 0.6 to 1.4%, respectively [30]. However, the values of these parameters can be significantly influenced by environmental and metabolic factors [31]. Post-harvest, keeping these parameters within the limits abovementioned ensures that the fruits keep the typical fresh and sweet taste over time. In this study, both the use of MAP and AL allowed for maintaining, unchanged, the basic characteristics after peeling during 21 days of storage, while CTR showed a marked qualitative decay after the 7th day of storage. These results suggest that both MAP and AL allow for slowing down the metabolic processes that underlie the qualitative decay of fruits. Also, the results obtained for the headspace confirm that CTR samples (without MAP and edible coating) produced a more intense metabolic activity due to respiration and transpiration processes that occurred after the peeling of the fruit [32,33]. Currently, there are no studies on edible coatings on Clementine segments; however, similar results were obtained by the application of edible coatings on whole citrus fruits [15] and Clementine fruits treated at pre-harvest with foliar application of Si-Ca and stored for 30 days [34].

The health, nutritional, and antioxidant properties of Clementine fruits have been recognized by many authors [2,35–39]. These properties are explained by the synergistic effect of phenolic compounds and organic acids (such as ascorbic, citric, and malic acid) present in fruits [23]. During storage, these compounds can be subjected to a decrease, depending on the storage conditions (packaging, temperature, lighting, and presence of oxygen and ethylene). In addition to physical and sensory characteristics, the purpose of this work was to investigate the effect of MAP and AL in preserving the bioactive compounds and antioxidant activity of ready-to-eat Clementine segments.

In our study, regarding TPC and TFC, significant differences were found among individual samples and during their shelf life. An initial increase in TPC and TFC was observed, followed by the maintenance of these constant values at the end of their shelf life. Several authors report that this phenomenon is mainly due to the activity of the enzyme PAL that, following the peeling operation, catalyzes the synthesis reactions of new phenolic compounds [1,40–42]. However, this rise could be followed by rapid decay if strategies to contain the oxidation of newly formed compounds do not fit. In this case, the combined effect of packaging, temperature, and MAP/coating resulted in a good result in terms of the maintenance of the compounds throughout the shelf life. A similar trend of TPC and TFC was observed in previous studies of citrus segments and juices [2].

The organic acid composition of the fruits is crucial for both the sensory and nutritional characteristics. In Clementine fruit, the major organic acids are citric, malic, ascorbic, and oxalic, respectively, and storage conditions can significantly affect their concentration [15, 23,43]. The obtained results denoted that MAP and AL maintained the levels of major organic acids present in the Clementine fruits over time. In particular, the level of ascorbic acid remained constant for all times of storage. In contrast, a drastic reduction of ascorbic acid was shown in the CTR sample. This is probably because only the packaging with barricaded material did not allow for slowing down the metabolic reactions that lead to its reduction. A previous study demonstrated that edible coatings and low temperature slow the activity of the enzyme aconitase and NADP-malic and gene expression levels, but they increase higher levels of citrate synthase and NAD-diseased dehydrogenase, resulting in reduced degradation of major acid present in the citrus fruit [44].

In summary, edible alginate-based coating preserves and improves the antioxidant capacity of the food because, during storage, there was an increase in TPC and TFC and maintenance of the initial level of organic acids. These results are significantly better than those obtained in CTR and MAP. In fact, at the end of shelf life, all the results related to antioxidant activity and nutritional properties (TPC, TFC, and organic acid) were significantly better in AL than MAP and CTR.

Finally, the microbiological analysis revealed that all the samples showed an absence of total bacteria, yeasts, and molds. This phenomenon is certainly the result of the processing conditions of the raw material (environment carefully sanitized), storage (low temperature, sealed tray, MAP, edible coating), and the constituent characteristics of the fruit, such as phenolic compounds and acidity that counteract/slow the proliferation of pathogenic and spoilage microorganisms.

## 5. Conclusions

An alginate-based edible coating favored the creation of a controlled environment with low oxygen stress, which resulted in a reduction in enzymatic activity and oxidation. In fact, coated samples (AL) showed better chemical and texture characteristics than CTR and MAP in terms of the highest content of total polyphenols and flavonoids, together with oxalic, citric, and ascorbic acids after 21 days of storage at 4 °C. In addition, the total acceptability scores suggest that coated samples retain their characteristics to a greater extent, even after 21 days of storage.

The use of an alginate-based edible coating may then be a better choice to preserve the quality of Clementine segments than packaging in a modified atmosphere. The use of natural and renewable resources to extend the shelf life of ready-to-eat fruits could represent a real opportunity for the fruit industries, for the consumer, and for the environment.

**Author Contributions:** Conceptualization, A.P.; methodology, A.P. and M.A.B.; validation, A.P. and A.D.B.; formal analysis, M.A.B.; data curation, A.P. and M.A.B.; writing—original draft preparation, M.A.B.; writing—review and editing, A.P. and A.D.B.; supervision, A.P. All authors have read and agreed to the published version of the manuscript.

**Funding:** This research received no external funding.

**Institutional Review Board Statement:** Not applicable.

**Informed Consent Statement:** Not applicable.

**Data Availability Statement:** Data are contained within the article.

**Conflicts of Interest:** The authors declare no conflict of interest.

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
