# Peer review of "Quality Evaluation of Ready-to-Eat Coated Clementine (Citrus x Clementina) Fruits"

_coatings, doi:10.3390/coatings13091562_

Round 1

Reviewer 1 Report

Quality evaluation of ready to eat coated citrus fruits

The effect of conventional and innovative preservation treatments to extend the shelf- life of ready-to-eat Clementine (Citrus x Clementina) segments were compared with the aim to find an environmentally and friendly quality preservation strategy and to meet the needs of the final consumer in terms of practicality use and food safety.

Comments:

The subject of the manuscript is interesting and the manuscript is also well written. However, some issues should be considered before publication.

1.     Why the authors did not use any multivariate data analysis method in this work? Multivariate data analysis methods such as “principal component analysis (PCA)” reflects a more comprehensive view of the system that provides more information about the relationship between the samples. Additionally, it can be used for determination of the most effective parameters in the quality of the final coated products.

2.     The authors did not mention the previous activities which have been performed in this field such as:

Evaluation of physicochemical properties of film-based alginate for food packing applications, M Azucena Castro-Yobal, A Contreras-Oliva… - e-Polymers, 2021

Characterization of sodium alginate-based films incorporated with thymol for fresh-cut apple packaging, J Chen, A Wu, M Yang, Y Ge, P Pristijono, J Li, B Xu… - Food Control, 2021 – Elsevier

Development of active packaging film from sodium alginate/carboxymethyl cellulose containing shallot waste extracts for anti-browning of fresh-cut produce,

P Thivya, YK Bhosale, S Anandakumar, V Hema… - International Journal of …, 2021 – Elsevier

Based on the above-mentioned issues, I think the manuscript could be considered for publication after a major revision.

Quality evaluation of ready to eat coated citrus fruits

The effect of conventional and innovative preservation treatments to extend the shelf- life of ready-to-eat Clementine (Citrus x Clementina) segments were compared with the aim to find an environmentally and friendly quality preservation strategy and to meet the needs of the final consumer in terms of practicality use and food safety.

Comments:

The subject of the manuscript is interesting and the manuscript is also well written. However, some issues should be considered before publication.

1.     Why the authors did not use any multivariate data analysis method in this work? Multivariate data analysis methods such as “principal component analysis (PCA)” reflects a more comprehensive view of the system that provides more information about the relationship between the samples. Additionally, it can be used for determination of the most effective parameters in the quality of the final coated products.

2.     The authors did not mention the previous activities which have been performed in this field such as:

Evaluation of physicochemical properties of film-based alginate for food packing applications, M Azucena Castro-Yobal, A Contreras-Oliva… - e-Polymers, 2021

Characterization of sodium alginate-based films incorporated with thymol for fresh-cut apple packaging, J Chen, A Wu, M Yang, Y Ge, P Pristijono, J Li, B Xu… - Food Control, 2021 – Elsevier

Development of active packaging film from sodium alginate/carboxymethyl cellulose containing shallot waste extracts for anti-browning of fresh-cut produce,

P Thivya, YK Bhosale, S Anandakumar, V Hema… - International Journal of …, 2021 – Elsevier

Based on the above-mentioned issues, I think the manuscript could be considered for publication after a major revision.

Author Response

you can see the replies in the attached file

Reviewer 2 Report

This manuscript (MS) aims to find an environmentally and friendly quality preservation strategy and to meet the needs of the final consumer in terms of practicality use and food safety. The MS was well designed and written, the findings provides well references for the post-harvest of fruits to fast-eating in the market. While some minor concerns need to be improved as follow:

1. The Standard errors in the Tables 1 to 5 need to be added.

2. All the data were shown in Tables, can some key data (e.g., bioactive compounds) be shown as Figures?

3. Some sentences are too long for readers to follow, such as the sentence in lines 8-11. Thus, they can be revised to easily read.

4. The Conclusion section should be concised to show the main findings of this MS.

5. The representative morphological character of the citrus fruits is suggested to be provided in the text.

Author Response

(The authors gave the same response as above.)

Reviewer 3 Report

This paper needs a major revision before publication. I have listed a few comments that need to be addressed:

1.       Add more concrete results in abstract.

2.       Improve the introduction part with more background about the work.

3.       What is the novelty of this research work that should be clearly discussed at the end of introduction? Also cite recent review and research article on this topic in introduction.

4.       Write the full form once when mentioning for the first instance.

5.       Add reference in methodology section.

6.       Add a schematic diagram to show the overall work

7.       Add apparent image of the packaging.

8.       Add more insights into the shelf life of the coated fruits.

9.       The integration of the results from different parameters should be improved carefully.

10.    Discussion part could be better, improve it.

11.    Conclusion could be better, improve it.

12.    Add recent References.

13.    Also, carefully revise the typos and linguistic errors to make the manuscript error free.

Author Response

(The authors gave the same response as above.)

Reviewer 4 Report

After reviewing your work, I would like to offer a series of recommendations aimed at further improving the quality and impact of your manuscript.

1. In order to provide a succinct and clear overview of your paper, I recommend that you explicitly state the main objective within the abstract section. This will enable readers to swiftly grasp the fundamental purpose of your research and its potential implications.

2. To enhance the coherence of your manuscript, consider outlining the motivation and contributions of your work in a point-wise manner within the introduction section. This structure will allow readers to efficiently discern the significance of your research and the unique insights it offers to the field.

3. In order to facilitate the reader's understanding of the logical progression of your paper, I suggest incorporating brief introductory paragraphs at the outset of each major section. These introductory paragraphs will serve to contextualize the content and aid in transitioning smoothly between sections.

4. To ensure the academic rigor of your paper, it is imperative that all cited references within the literature review are accurately presented. Please meticulously verify and meticulously format your citations, adhering to the prescribed citation style.

5. In the interest of providing a comprehensive analysis, I recommend introducing a detailed discussion at the inception of section 3. This discussion should explicitly elucidate the ways in which your obtained results surpass those of existing methodologies. By thoroughly highlighting the superior attributes of your work, you will fortify the originality and potency of your research.

6. To enhance the visual comprehension of your research, I propose incorporating graphical presentations within section 4. These visuals can effectively convey complex information and contribute to a more dynamic and engaging reader experience.

7. I suggest that you revisit and refine your conclusion section. In addition to succinctly summarizing your findings, consider appending a subsection that delineates the future directions in which this research could be extended. Additionally, address the limitations of your work in a transparent manner, acknowledging areas where further exploration is warranted.

8. In order to underscore the contemporaneity of your research, it is advisable to include references that have been published after the year 2022. This practice will not only inform the reader of the latest advancements in the field but also underscore the relevance of your work within the current academic landscape.

Extensive editing of English language required

Author Response

(The authors gave the same response as above.)

Round 2

Reviewer 4 Report

Accepted

Minor changes.